# Pharmacologic Management of End-of-Life Delirium: Translating Evidence into Practice

**DOI:** 10.3390/cancers16112045

**Published:** 2024-05-28

**Authors:** David Hui, Shao-Yi Cheng, Carlos Eduardo Paiva

**Affiliations:** 1Department of Palliative, Rehabilitation and Integrative Medicine, University of Texas MD Anderson Cancer Centre, Houston, TX 77030, USA; 2Department of Family Medicine, College of Medicine and Hospital, National Taiwan University, Taipei 10617, Taiwan; scheng2140@gmail.com; 3Department of Clinical Oncology, Barretos Cancer Hospital, Barretos 1331, SP, Brazil; caredupai@gmail.com

**Keywords:** antipsychotic agents, benzodiazepines, delirium, death, drug therapy, palliative care, neoplasms, randomized controlled trial, terminally ill

## Abstract

**Simple Summary:**

In the last days and weeks of life, many patients with cancer develop delirium with fluctuating confusion, altered levels of consciousness, hallucinations, restlessness, and, sometimes, agitation, which fluctuate over time. This condition can be very distressing for the patient and their family. In this article, the authors reviewed different treatments for delirium in patients receiving palliative care. Previous studies have shown that certain medications, such as neuroleptics and benzodiazepines, may be helpful in controlling the terminal agitation associated with end-of-life delirium. However, further research is needed to confirm these findings and find new treatment options to address this challenging issue.

**Abstract:**

End-of-life delirium affects a vast majority of patients before death. It is highly distressing and often associated with restlessness or agitation. Unlike delirium in other settings, it is considered irreversible, and non-pharmacologic measures may be less feasible. The objective of this review is to provide an in-depth discussion of the clinical trials on delirium in the palliative care setting, with a particular focus on studies investigating pharmacologic interventions for end-of-life delirium. To date, only six randomized trials have examined pharmacologic options in palliative care populations, and only two have focused on end-of-life delirium. These studies suggest that neuroleptics and benzodiazepines may be beneficial for the control of the terminal restlessness or agitation associated with end-of-life delirium. However, existing studies have significant methodologic limitations. Further studies are needed to confirm these findings and examine novel therapeutic options to manage this distressing syndrome.

## 1. Introduction

Delirium is characterized by an acute disturbance of consciousness with altered awareness and attention, decreased cognition, and perceptual abnormalities that fluctuate over time [1]. It is the most common neuropsychiatric syndrome in patients with advanced cancer [2,3]. Delirium is present in 10–31% of hospitalized patients upon admission [4], 31% of patients admitted to intensive care units [5], 9–57% of patients seen by inpatient palliative care teams, 6–74% of patients admitted to palliative care units, and 42–88% of patients before death [6]. Delirium in different settings and patient populations varies widely in terms of etiologies, prevalence, clinical courses, and outcomes.

End-of-life delirium, also known as terminal delirium, is an acute brain dysfunction that occurs in the last weeks-to-days of life and is considered to be irreversible [7,8]. End-of-life delirium may be nearly universal as brain function deteriorates as part of the dying process. End-of-life delirium should be distinguished from delirium in patients who are “terminally ill”. The later generally indicates a life expectancy of 6 months or less [9], and these patients may have a better chance of delirium recovery, although some may have or would eventually develop end-of-life delirium [10]. A majority of the studies on end-of-life delirium have been conducted in the advanced cancer setting. There is insufficient evidence to inform us if end-of-life delirium differs in non-malignant diseases. Much of the literature is derived from acute palliative care units, where patients are admitted for severe physical and/or psychosocial distress, with the need of intensive end-of-life care and/or complex discharge coordination. Although not typically a requirement, many of these patients have a prognosis in terms of days to weeks of survival, and end-of-life delirium is common. Because of the extremely short life expectancy of this patient population and clinical course, the goals of care for end-of-life delirium differs from other populations—palliation of delirium symptoms, finding a balance between communication and the need for sedation, and support for caregivers are paramount (Table 1) [11].

End-of-life delirium has a negative impact on almost every aspect of patient care, including symptom assessment, patient–clinician communication, and decision making [7]. In addition, delirium is associated with prolonged hospitalizations, higher healthcare costs, and increased mortality [2,12,13,14]. Approximately 50–70% of patients with delirium have hyperactive or mixed subtypes that are characterized by agitation and often associated with hallucinations, delusions, and hypervigilance [15]. Agitation, which ranges from restlessness to aggressive violent behavior, can pose a safety risk for patients, caregivers, and healthcare professionals and can be highly distressing to all involved [16,17]. In a prospective study examining agitation in delirium, the mean delirium-related distress level was 3.2 out of 4 for patients (with 4 being the most severe), 3.75 out of 4 for caregivers, and 3.1 out of 4 for nurses [18,19]. Patients’ agitation and the associated symptoms may impede their communication with their families and hinder their participation in treatment decisions, counseling, and symptom assessment [20]. A systematic review that included 33 studies examining caregivers’ perspective concluded that “high levels of distress are experienced by caregivers of patients with delirium” in the palliative care setting [21]. Patients with poorly controlled end-of-life delirium were found to have a lower quality of end-of-life care [22].

The management of end-of-life delirium is controversial because of the lack of mechanistic studies to understand its pathophysiology and the paucity of clinical trials in this patient population to inform practice. Thus, much of the management of end-of-life delirium has been based on studies outside of the palliative care setting that may not be applicable to this population. The management of delirium generally involves (1) identifying and removing any reversible causes if appropriate [23,24], (2) offering non-pharmacologic interventions [25,26], and (3) providing pharmacologic treatments for palliation [3,27]. Unfortunately, the first two options may have a lesser role in end-of-life delirium. Specifically, the diagnosis of end-of-life delirium indicates that it is considered irreversible, either because the limited prognosis makes it inappropriate to investigate or treat any underlying causes, or the delirium has not improved despite such measures. Multicomponent non-pharmacological interventions, which typically include orientation cues, hearing or visual aids, hydration protocols, and early ambulation have been found in systematic reviews to prevent delirium in hospitalized patients [26]. However, a vast majority of these studies were conducted outside of the palliative care setting, and the interventions are often not feasible in patients in their last days of life due to extreme weakness and unresponsiveness. Only two controlled studies in palliative care populations have been reported, and both were focused on preventing (instead of treating) delirium—a non-randomized study found a high incidence of delirium in the intervention group [28], and another reported low adherence to non-pharmacological interventions [29]. The impact of non-pharmacologic measures on the caregivers of patients with delirium, such as window light and music, remains a subject for further investigation. Education of caregivers of the nature history, manifestations, and treatment options for end-of-life delirium, exploration of their values, and setting realistic expectations are essential.

Because of the irreversibility of end-of-life delirium, the effective palliation of delirium symptoms is of paramount importance. Given that non-pharmacologic approaches have limited feasibility and evidence to support their use in actively dying patients and the urgency of symptom control, many clinicians would consider pharmacologic approaches for patients with restlessness or agitation [30]. In 2020, a Cochrane Database systematic review on delirium in terminally ill adults (not specifically end-of-life delirium) was published [31]. This study included four randomized trials with 399 participants [32,33,34,35] and concluded that there was “no high-quality evidence to support or refute the use of drug therapy for delirium symptoms in terminally ill adults”. Since then, two more randomized trials have been published in the palliative care setting [36,37]. Despite some methodologic limitations, these clinical trials provide useful data to inform clinical practice and future delirium trial design (Table 2). The focus of this review is to provide an in-depth discussion of the clinical trials on delirium in the palliative care setting, with a particular focus on studies investigating pharmacologic interventions for end-of-life delirium.

## 2. Neuroleptics for Delirium in the Palliative Care Setting

To date, four randomized clinical trials have examined first-line pharmacologic therapy for delirium in the palliative care setting. These studies included patients with both hyperactive and hypoactive features and assessed delirium severity or a composite of delirium-related symptoms as the key outcomes.

Breitbart et al., from the United States, conducted a double-blind, randomized trial to compare haloperidol (n = 11), chlorpromazine (n = 13), and lorazepam (n = 6) in 30 terminally ill patients with acquired immunodeficiency syndrome and delirium in 1996 [34]. Although the patients were not seen by palliative care, this was the only clinical trial at the time focusing on patients with a terminal illness and, as such, informed clinical practice. Upon enrollment, the patients were monitored with the Delirium Rating Scale (DRS) hourly and were started on oral or intramuscular study medications when the DRS was 13 or higher. The medications were actively titrated every hour until the patient was asleep, calm, and not hallucinating, or the DRS score was less than 13. The total dose requirement in the first 24 h was 2.8 mg in the haloperidol group, 50 mg in the chlorpromazine group, and 3 mg in the lorazepam group. The patients then received 50% of the total first day dose twice daily between day 2 and day 6. The DRS had a significant within-group reduction in the haloperidol group (from 20.5 at the baseline to 12.5 on day 2 and 11.6 at the end of treatment, *p* < 0.001) and chlorpromazine group (from 20.6 at the baseline to 12.1 on day 2 and 11.9 at the end of treatment, *p* < 0.001) but not lorazepam (from 18.3 at the baseline to 17.3 on day 2 and 17.0 at the end of treatment, *p* < 0.63) [34]. There was no significant between-group difference. The Mini-Mental State scores showed similar findings. The lorazepam group was terminated after six patients because of substantial treatment limiting adverse events, including over-sedation, disinhibition, ataxia, and worsening confusion. While this study was underpowered for between-group comparison, it suggested that neuroleptics were associated with some improvement in delirium in this population. However, the lack of a placebo control meant that it was not possible to know if any improvement was a result of the neuroleptics or the natural history of delirium. Nevertheless, many clinicians have been prescribing neuroleptics and avoiding lorazepam based on the findings in this study.

In 2008, Lin et al., from Taiwan, reported their open-label randomized trial assessing the within-group efficacy of oral haloperidol (n = 14) and olanzapine (n = 16) for delirium in patients with advanced cancer and admitted to a hospice and palliative care center, representing the first clinical trial specifically in the palliative care setting [32]. Fourteen patients received oral haloperidol 5 mg daily and sixteen received oral olanzapine 5 mg daily; the doses were titrated over time. The DRS, the key outcome, improved from 16.5 at the baseline to 11.9 on day 1, 13.0 on day 2, and 12.3 on day 7 in the haloperidol group (*p* = 0.04). The olanzapine group also had a similar level of improvement (17.6 at the baseline to 14.3 on day 1, 14.9 on day 2, and 10.6 on day 7, *p* = 0.04) [32]. No significant difference was found between the two groups. The authors concluded that both haloperidol and olanzapine were efficacious in reducing delirium symptoms in patients with advanced cancer. However, the small sample size and lack of a placebo control were important limitations.

In 2017, Agar and colleagues, from Australia, completed a multicenter, double-blind, randomized trial comparing oral risperidone, haloperidol, and placebo in delirious patients receiving palliative care [33]. This delirium trial marked the first time a placebo was examined in the palliative care setting. The primary outcome was a change in a composite score consisting of three delirium symptoms (i.e., hallucinations, communications, and behavior) in the Nursing Delirium Screening Scale (total score out of six, with higher scores indicating worse symptoms). The key secondary outcomes were delirium severity, the use of midazolam, extrapyramidal effects, sedation, and survival. A total of 247 patients were included, with 82 in the risperidone group, 81 in the haloperidol group, and 84 in the placebo group. The doses were low, with both risperidone and haloperidol starting at 0.5 mg every 12 h. Subcutaneous midazolam 2.5 mg was available as a rescue every 2 h, as needed. Compared to the placebo, both risperidone (mean difference [95% CI]: 0.48 [0.09–0.86]; *p* = 0.02) and haloperidol (0.24 [0.06–0.042]; *p* = 0.009) had significantly worse delirium symptom scores [33]. Both neuroleptics required a more frequent use of rescue midazolam compared to the placebo (day 1: 34.7% vs. 17.3%; *p* = 0.007 33.1% vs. 16.8%; *p* = 0.01 and 29.6% vs. 13.6% on day 3). In addition, the extrapyramidal effects were higher in both the risperidone (mean difference [95% CI]: 0.73 [0.09–1.37]; *p* = 0.03) and haloperidol groups (0.79 [0.17–1.41]; *p* = 0.01) compared to the placebo. The median overall survival was in favor of the placebo (risperidone 17 days, haloperidol 16 days, and placebo 26 days; *p* = 0.01). The strengths of this study included the large sample size, the multicenter design, and the inclusion of a placebo group as the control. However, the primary outcome was not validated, and it was unclear if any difference was clinically meaningful [38]. Further complicating interpretation is the fact that this study included patients with milder delirium (median MDAS 14–15), and approximately 20% of the patients had an underlying cognitive impairment, meaning that the adverse effects may have been more pronounced in this patient group. Moreover, the medication doses were low. Based on these findings, some investigators call for the avoidance of medication to focus on non-pharmacologic measures. However, it should be noted that the placebo group had midazolam, and the non-pharmacologic measures were not fully standardized or documented in this clinical trial.

More recently, van der Vorst et al., from the Netherlands, completed a multicenter single-blind randomized trial comparing haloperidol and olanzapine in 100 patients with advanced cancer enrolled from medical oncology wards or high-care hospice facilities [37]. Forty-nine patients received oral or subcutaneous haloperidol, and forty-nine received oral or intramuscular olanzapine. Only the research staff conducting the assessments were blinded; the clinical team and the patients were unblinded. The starting dose was 1 mg for haloperidol and 5 mg for olanzapine; the doses were titrated every 40 min if the Delirium Observation Scale score was greater than or equal to 3. After the first day, the patients received 50% of the total 24 h dose as maintenance. In the first 24 h, the median dose was 2.5 mg for haloperidol and 8.8 mg for olanzapine. The primary outcome was the delirium response rate on days 1 to 7, defined as a DRS-R-98 severity score < 15.25 or a reduction of ≥4.5 points in the total score. The investigators estimated that 200 patients were required to compare groups with a 25% difference in their delirium response rate. However, this study was stopped early, after 50% enrollment, due to futility. The delirium response rate was 57% for haloperidol and 45% for olanzapine (*p* = 0.20) [37]. The mean time to the treatment response was 2.8 days for haloperidol and 4.5 days for olanzapine (*p* = 0.23). The authors concluded that both neuroleptics were equally effective for the management of delirium. Unfortunately, this study also had several important limitations, including heterogenous patient populations, no masking of patients and clinicians, non-equivalent doses, a lack of a placebo control, and an unclear survival.

Taken together, three randomized trials that did not include a placebo group showed within-group improvement over time with neuroleptics; however, the only placebo-controlled trial suggested that neuroleptics were not only ineffective but also potentially harmful. Moreover, the Breitbart study suggested that lorazepam may be overly sedating, while the Agar study highlighted midazolam’s role as a rescue medication. The confusing evidence resulted in significant debate and variations in clinical practice [30,31,39,40,41,42].

Of note, several large, double-blind, placebo-controlled randomized trials of haloperidol and other neuroleptics outside of the palliative care setting revealed that neuroleptics had no significant impact on delirium severity or duration [43,44,45,46,47].

## 3. Benzodiazepines for Persistent Agitated End-of-Life Delirium

Benzodiazepines are considered the drugs of choice for patients with delirium tremens [48]; however, outside of this setting, there has been much debate over whether benzodiazepines should be given to patients with delirium [49,50]. Many clinicians avoid the use of benzodiazepines because of their well-established adverse-event profile, including the precipitation of delirium, over-sedation, memory loss, falls, and a possibly worsened survival [34,51]. At the same time, some clinicians have observed that benzodiazepines can reduce agitation quickly and reliably. To address this question, we conducted the Medications for Agitated Delirium (MAD) study, a single-center, double-blind, randomized clinical trial to compare the effect of a single dose of haloperidol 2 mg plus lorazepam 2 mg to haloperidol 2 mg plus placebo for an episode of agitation in patients with terminal delirium admitted to our acute palliative care unit [35]. Because of the concerns regarding harm related to lorazepam at the time of study design, this study only examined a single dose of the study medication, and both groups were given haloperidol as a rescue, which is considered the standard of care.

We included adult patients with advanced cancer admitted to our acute palliative care unit who had been diagnosed with delirium, had a history of agitation, and were treated with scheduled haloperidol doses ranging from 1 to 8 mg/day [35]. After surrogate consent was obtained, the patients started on the open-label phase, during which their delirium regimen was standardized to haloperidol 2 mg every 6 h and haloperidol 1 mg every 1 h, as needed. They were also randomized in a 1:1 ratio to either haloperidol plus lorazepam or haloperidol plus placebo, which was given when they became more restless (Richmond Agitation Sedation Scale [RASS] Score 1+ or higher), despite the open-label haloperidol. The primary outcome was the change in RASS score from the start of blinded medication administration to 8 h after.

We enrolled 90 patients in total, with 47 patients assigned to the lorazepam-plus-haloperidol group and 43 to the lorazepam-plus-placebo group [35]. Twenty-nine patients in each group started the blinded phase, and twenty-six patients in each group completed 8 h of observation. The patients on the haloperidol-plus-lorazepam group had a significantly greater reduction in their RASS score within the first 30 min compared to the haloperidol-plus-placebo group (mean [95% CI]: −3.6 [−4.3 to −2.9] vs. −1.6 [−2.2 to −1.0]; *p* < 0.001), and this effect was sustained over the next 8 h (−4.1 [−4.8 to −3.4] vs. −2.3 [−2.9 to −1.6]; *p* < 0.001). Over the 8 h observation period, the patients on the haloperidol + lorazepam group also a had fewer number of breakthrough restlessness episodes (28% vs. 76%, *p* < 0.001), received fewer doses of rescue medications (mean 1.0 vs. 2.0, *p* = 0.009), and were perceived by both caregivers and bedside nurses to be more comfortable. The adverse events (mostly hypokinesia and akathisia) were similar between the two groups, and the overall survival did not differ significantly (68 h vs. 73 h, *p* = 0.56).

Taken together, the combination of haloperidol plus lorazepam was more effective in reducing agitation than haloperidol alone. However, this study was limited to a single rescue dose, was only conducted in a single center, had a short observation period, enrolled a relatively small number of patients, and lacked a placebo-only group. This raises more questions: Would lorazepam alone be as effective as lorazepam plus haloperidol? Would a more proactive approach with scheduled medications be preferred? What if lorazepam was given alone without haloperidol? Is the reduction in the RASS score after the administration of haloperidol a direct effect of medication or a natural fluctuation of delirium? Would surrogate decision makers be open to enrolling their families into a randomized trial with a placebo-only group? As an attempt to answer some of these questions, our research team designed the RECORD (‘Rotation, Escalation, Combination, Or Reduction for Delirium) study to compare scheduled haloperidol, lorazepam, lorazepam plus haloperidol, and placebo (NCT03743649) (Figure 1). This multicenter, double-blind, randomized trial enrolled 111 patients with advanced cancer from the MD Anderson Cancer Center, the Virginia Commonwealth University, and the Taiwan National University Hospital. It has just completed enrollment, and the results are expected in 2024.

## 4. Neuroleptic Rotation and Combination for Persistent Agitated End-of-Life Delirium

In addition to benzodiazepines, clinicians may manage patients with agitated delirium who have not responded to low-dose haloperidol by increasing the dose of haloperidol, switching to another neuroleptic, or adding the other neuroleptic to haloperidol. Several medications, such as chlorpromazine, levomepromazine, quetiapine, risperidone, olanzapine, and bloanserin, have been reported in retrospective and/or prospective series to reduce agitation [39,52,53,54,55,56]. Medications that are available parenterally, such as haloperidol and chlorpromazine, are preferred in the agitated end-of-life delirium setting because of the urgency of the clinical situation, swallowing difficulties, and an inability to cooperate in this population.

The Chlorpromazine and Haloperidol for Agitated Delirium (CHAD) study is a single-center, double-blind, double-dummy randomized clinical trial to examine the effect of haloperidol, chlorpromazine, and the combination of haloperidol and chlorpromazine on terminal agitation [36]. At the time of study design, we decided not to include a placebo-only control group because of patient safety concerns. Patients admitted to an acute palliative care unit were eligible for this study if they had advanced cancer, a clinical diagnosis of hyperactive or mixed delirium, persistent restlessness, or agitation despite low-dose haloperidol of 1–8 mg/day, and no contraindications to neuroleptics. Upon enrollment, the patients were started on the open-label phase, during which their treatment was standardized to haloperidol 2 mg intravenously every six hours and 2 mg intravenously every one hour, as needed. The patients who continued to develop restlessness or agitation proceeded to the blinded phase, during which they were given the scheduled study medication every 4 h. Specifically, the dose for the haloperidol group was 2 mg; the dose for the chlorpromazine group was 25 mg; and the dose for the combination group was haloperidol 1 mg and chlorpromazine 12.5 mg. The doses for all three groups were designed to be equivalent to facilitate comparison. Rescue doses identical to the scheduled dose in the respective arms may have been given every 1 h, as needed.

The primary outcome was a within-group change in the RASS score after 24 h of blinded medications [36]. The secondary outcomes included rescue medication use, patient comfort as perceived by caregivers and nurses, delirium-related distress (Delirium Experience Questionnaire), delirium severity (Memorial Delirium Assessment Scale), symptom burden (Edmonton Symptom Assessment System), and adverse events (Common Toxicity Criteria for Adverse Events, version 4.03, Udvalg for Kliniske Undersogelser Rating Scale). This study enrolled 68 patients, with 23 allocated to the haloperidol group, 22 to the chlorpromazine group, and 23 to the combination group. However, some patients did not proceed to the blinded phase because they had no more agitation after starting open-label haloperidol or died before blinded medication could be initiated. The primary analysis included 15 patients in the haloperidol group, 16 patients in the chlorpromazine group, and 14 patients in the combination group.

The RASS score significantly decreased in all three groups within 30 min of blinded medication administration compared to the baseline (mean [95% CI]: haloperidol group −2.6 [−3.6 to −1.6]; chlorpromazine group −2.4 [−3.0 to −1.8]; and combination group −2.1 [−3.0 to −1.3]). After 24 h, the changes in the RASS score were −3.6 (−5.0 to −2.2) for haloperidol, −3.3 (−4.4 to −2.2) for chlorpromazine, and −3.0 (−4.6 to −1.4) for the combination [36]. This study was not powered for between-group comparison, and we did not detect a statistically significant difference among the three groups (*p* = 0.71). Thus, all three strategies appeared to be able to reduce agitation to a similar extent; however, the absence of a placebo-only control group limited our interpretation.

A number of important secondary outcomes warrant discussion. Specifically, 73%, 19%, and 50% of patients in the haloperidol group, chlorpromazine group, and combination group developed breakthrough restlessness (RASS ≥ +1) within the first 4 h, respectively (*p* = 0.009). These proportions increased to 80%, 25%, and 44% after 8 h (*p* = 0.01) and 80%, 50%, and 64% after 24 h (*p* = 0.12). In addition, patients in the haloperidol group and combination group required more rescue neuroleptics than those in the chlorpromazine group (median haloperidol equivalent daily dose: 4 mg vs. 6 mg vs. 2 mg; *p* = 0.09); similarly, more patients in the haloperidol group and combination group required study medication dose escalation (27% vs. 50% vs. 6%, *p* = 0.03). The perceived patient comfort by both caregivers (haloperidol 62%, chlorpromazine 71% and combination 60%; *p* = 0.83) and bedside nurses (haloperidol 64%, chlorpromazine 75% and combination 64%; *p* = 0.82) was also in favor of chlorpromazine, albeit non-statistically significant.

Taken together, the CHAD data suggest that chlorpromazine may be a more proactive approach to managing agitation than haloperidol [36]. Although the patients achieved similar RASS scores in all three groups after 24 h, those in the haloperidol and combination groups had more breakthrough episodes and required active efforts (rescue medications, dose escalation, and extra medication outside of the study medications) to reduce their agitation. Surprisingly, the combination group, administered half of the haloperidol dose and half of the chlorpromazine dose, performed the worst. This may be a result of the lower doses of both medications. This randomized trial had several limitations, including the lack of a placebo-only control group and the small number of enrolled patients. Larger studies are needed to confirm our findings.

There are several ongoing studies to examine the use of neuroleptics in end-of-life delirium. One randomized trial (HALOLAN trial, NCT04750395) compared oral transmucosal haloperidol 2.5 mg to oral transmucosal olanzapine 5 mg for the management of end-of-life delirium at home over a 72 h treatment period [57]. The HALO trial (NCT04833023) was a multicenter, open-label, randomized controlled trial that compared the use of oral haloperidol (escalating doses from 1 mg to 6 mg/24 h) versus oral olanzapine (2.5 to 15 mg/24 h) in patients with advanced cancer or end-stage organ disease experiencing hyperactive delirium. A rescue dose of midazolam 2 mg every 2 h, as needed, was available in both arms.

Dexmedetomidine, a selective α2 adrenergic receptor agonist, has emerged as a promising agent for managing hyperactive medical delirium. A meta-analysis has indicated its superiority over other medications, such as haloperidol, in shortening delirium duration among ICU patients [58]. However, in patients under palliative care, the drug has not yet been prospectively tested. An open-label, single-arm feasibility trial (NCT04824144) is currently underway in Canada to test the use of the subcutaneous continuous infusion of dexmedetomidine in hospitalized patients with end-of-life delirium. While commonly utilized in intensive care settings, its application in palliative care is not recommended based on current data, except within a research framework.

## 5. Clinical Practice

Based on our review of the literature, we propose the following approach to the management of end-of-life delirium (Figure 2). The diagnosis of end-of-life delirium requires the appropriate clinical context and careful assessment. Because of the challenges with prognostication [59,60], it may be difficult to know if a patient is in their last weeks of life and whether the patient’s delirium is truly irreversible. Unless the patient is clearly dying [61,62], it may be reasonable to look for and treat any potentially reversible causes [23,24]. Once the diagnosis of end-of-life delirium is made, it is important to educate caregivers on the irreversible nature of this syndrome, the various manifestations of delirium (e.g., restlessness, agitation, hallucinations, fluctuating level of consciousness, and inattention), and the process of dying [11,63]. Continual emotional support and discussions about the goals of care are essential.

Despite the absence of data on non-pharmacologic measures in the treatment of end-of-life delirium, selected measures, such as window light, orientation cues, and family photographs, may be considered where appropriate.

For patients with hypoactive end-of-life delirium, pharmacologic measures are not indicated. As it is not known which of these patients may develop mixed delirium, we typically suggest an order of haloperidol on an as-needed basis for terminal restlessness or agitation.

For patients with terminal restlessness in the context of end-of-life delirium, we recommend scheduling low-dose haloperidol (<8 mg/day) as a first-line treatment, coupled with a rescue up to every 1 h, as needed. For patients with persistent agitation despite the administration low-dose haloperidol, the second-line options include the escalation of the scheduled haloperidol dose (>8 mg/day), switching to another neuroleptic (e.g., chlorpromazine), or adding lorazepam to haloperidol as a rescue. Patients need to be monitored closely for adverse events (e.g., extrapyramidal side effects) and agitation control. For patients with refractory agitation, continuous deep palliation sedation (e.g., midazolam infusion) may be required after a thoughtful discussion with family [64].

## 6. Conclusions

End-of-life delirium affects a vast majority of patients before death. It is highly distressing and often associated with restlessness or agitation. Unlike delirium in other settings, it is considered irreversible, and non-pharmacologic measures may be less feasible. To date, only six randomized trials have examined pharmacologic options in palliative care populations, and only two have focused on end-of-life delirium. These studies suggest that neuroleptics and benzodiazepines may be beneficial for the control of the terminal restlessness or agitation associated with end-of-life delirium. However, existing studies have significant methodologic limitations. Further studies are needed to confirm these findings and examine novel therapeutic options to manage this distressing syndrome.

## Figures and Tables

**Figure 1 cancers-16-02045-f001:**
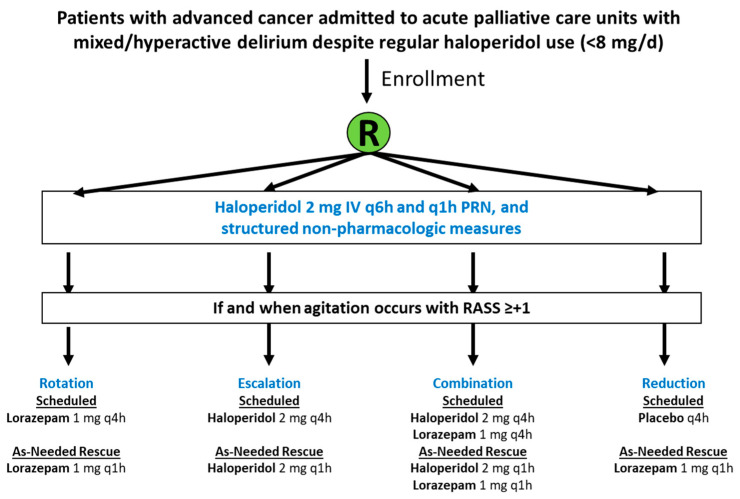
RECORD (Rotation, Escalation, Combination, Or Reduction for Delirium) study design.

**Figure 2 cancers-16-02045-f002:**
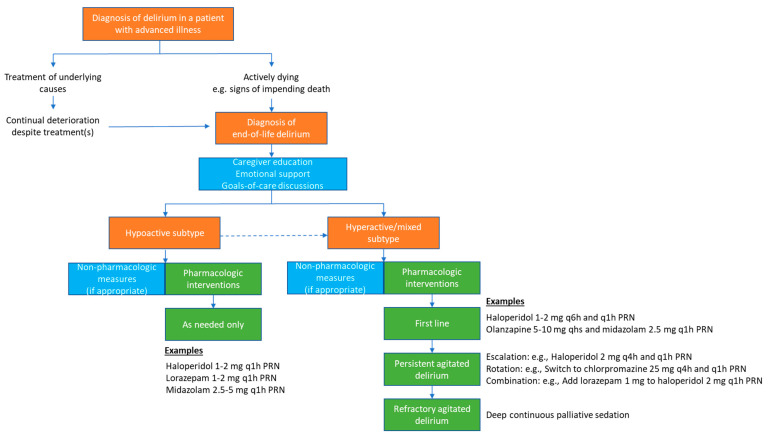
Management of end-of-life delirium.

**Table 1 cancers-16-02045-t001:** Differences between medical delirium and end-of-life delirium.

	Medical Delirium	End-of-Life Delirium
Patient population	Highly variable, typically months of prognosis (but could be much longer or shorter)	Relatively homogeneous. Typically, days of prognosis, maybe weeks. Patients may or may not have signs of impending death.
Main driver(s)	Medical complications, often multifactorial	The dying brain is the key driver.
Reversibility	Expected	Not expected.
Goals of treatment	Reduce delirium severity; reverse or shorten delirium	Palliation of symptoms such as agitation.
Treatment of reversible causes	Critical importance	May be attempted to rule out reversibility. May not be appropriate in patients with days of life expectancy.
Non-pharmacologic measures	Good evidence to support these interventions to prevent delirium in high-risk medical patients Inadequate evidence to support their use for the treatment of delirium	Lack of evidence to support their use for the prevention or treatment of delirium These may not be feasible in extremely sick patients; however, they may be reasonable to try if feasible given no risk and low cost.
Education of caregivers	Critical to set expectations; hope for reversal but deterioration is possible	Critical to set expectations; comfort as the key goal; need to discuss prognosis.
Pharmacologic treatments	Neuroleptics have no effect on delirium severity or duration	Neuroleptics and benzodiazepines may reduce terminal restlessness/agitation.

**Table 2 cancers-16-02045-t002:** Randomized clinical trials of delirium involving palliative care populations.

Author/Journal	Study Population/Setting	Design	Key Findings
Front Line Treatment of Medical Delirium in Palliative Care Settings			
Breitbart et al. *Am J Psych* 1996 [34]	30 patients with HIV First-line treatment for delirium Medically hospitalized patients Median survival not reported	Double-blind RCT Haloperidol vs. chlorpromazine vs. lorazepam Scheduled doses for 6 days Oral or intramuscular routes Primary outcome was not stated	Haloperidol and chlorpromazine showed within-group improvement in DRS by day 2, but not lorazepam. The chlorpromazine group also showed within-group improvement in the Mini-Mental State Exam by day 2.
Lin et al. *J Intern Med Taiwan* 2008 [32]	30 patients with advanced cancer First-line treatment for delirium; 70% had hyperactive/mixed delirium Single palliative care/hospice unit Median survival not reported	Open-label RCT Haloperidol vs. olanzapine Scheduled doses, duration unknown Oral route Primary outcome was not stated	Haloperidol and olanzapine both showed within-group improvement in the DRS and Global Impression Severity scores by day 7 but did not differ significantly.
Agar et al. *JAMA Intern Med* 2017 [33]	247 patients with advanced illnesses (88% advanced cancer) First-line treatment for delirium Multicenter inpatient palliative care and hospice units Median survival ~3 weeks	Double-blind RCT Haloperidol vs. risperidone vs. placebo Scheduled doses for 72 h Oral route Primary outcome was a composite of three NuDESC items on day 3	The haloperidol and risperidone groups had significantly worse NuDESC scores than the placebo and more extrapyramidal effects. The haloperidol group also had a poorer survival than the placebo.
Van der Vorst et al. *Oncologist* 2020 [37]	98 patients with advanced cancer First-line treatment for delirium; 70% had hyperactive/mixed delirium Multicenter medical oncology ward and high-care hospice units Median survival not reported	Single-blind RCT Haloperidol vs. olanzapine Scheduled doses until resolution Oral or subcutaneous routes Primary outcome was delirium response rate by DRS-R-98 on days 1–7	Haloperidol and olanzapine did not differ significantly in the response rate (57% vs. 45%, *p* = 0.23) and the time to response (2.8 d vs. 4.5 d, *p* = 0.18).
Treatment of Persistent Agitated End-of-Life Delirium in Palliative Care Settings			
Hui et al. *JAMA* 2017 [35]	58 patients with advanced cancer Persistent agitated delirium Single palliative care unit Median survival ~3 days	Double-blind RCT Haloperidol plus lorazepam vs. haloperidol plus placebo Single rescue dose Intravenous route Primary outcome was a change in RASS after 8 h	The addition of lorazepam resulted in a significant reduction in RASS after 8 h (−4.1 vs. −2.3, *p* < 0.001), fewer breakthrough restlessness episodes (28% vs. 76%, *p* < 0.001), fewer doses of rescue medications (1.0 vs. 2.0, *p* = 0.009), and greater comfort as perceived by both caregivers and bedside nurses.
Hui et al. *Lancet Oncol* 2020 [36]	45 patients with advanced cancer admitted to an acute palliative care unit Persistent agitated delirium Singe palliative care unit Median survival ~3 days	Double-blind RCT Haloperidol vs. chlorpromazine vs. combination Scheduled and rescue doses until discharge/death Intravenous medications Primary outcome was a change in RASS after 24 h	The haloperidol, chlorpromazine, and combination groups had a similar reduction in RASS after 24 h; however, the chlorpromazine group required fewer rescue doses and was less likely to have breakthrough restlessness than the other two groups.

Abbreviations: HIV, Human Immunodeficiency Virus; NuDesc, Nursing Delirium Screening Scale; RASS, Richmond Agitation Sedation Scale; and RCT, randomized controlled trial.

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
