# Peer review of "Pharmacologic Management of End-of-Life Delirium: Translating Evidence into Practice"

_cancers, 2024, doi:10.3390/cancers16112045_

Round 1

Reviewer 1 Report

Comments and Suggestions for Authors

While you seem to have set off with delirium in terminal cancer patients, midway, the narrayive peters towards delirium in a person dying fromany cause. 

If the intention is to focus on delirium in the dying patient, irrespective of the underlying disease, why point out to cancer alone as a possible cause?

It is also important to clearly state the ways and means of arriving at the diagnosis of delirium and, differentiating it from deliriumfrom other causes. What are the correctable causes of delirium in a dying patient? Does it matter whether the causes are deciphered and corrected?

The armamentarium suggested for management does not move beyond haloperidol and lorazepam. 

Comments on the Quality of English Language

Kindly refrain from quaint English expressions 

Author Response

While you seem to have set off with delirium in terminal cancer patients, midway, the narrayive peters towards delirium in a person dying from any cause. 

If the intention is to focus on delirium in the dying patient, irrespective of the underlying disease, why point out to cancer alone as a possible cause?

REPLY: Thank you for your comment.  Based on your feedback, we have now added the following: “Many of the studies on end-of-life delirium were conducted in the advanced cancer setting. There is insufficient evidence to inform us if end-of-life delirium differs in non-malignant diseases”.  We also discussed end-of-life delirium in the context of cancer given this is the Cancers journal.

It is also important to clearly state the ways and means of arriving at the diagnosis of delirium and, differentiating it from delirium from other causes. What are the correctable causes of delirium in a dying patient? Does it matter whether the causes are deciphered and corrected?

REPLY: Thank you for your comment.  We have now clarified that “the diagnosis of end-of-life delirium indicates that it is considered irreversible, either because the limited prognosis makes it inappropriate to investigate or treat any underlying causes, or the delirium has not improved despite such measures.”

The armamentarium suggested for management does not move beyond haloperidol and lorazepam. 

REPLY: Thank you for your comment.  This review focused on existing literature on end-of-life delirium which is quite limited.  Among the medications discussed were haloperidol, olanzapine, chlorpromazine, lorazepam and midazolam.  We have now revised the figure to list more options.

Kindly refrain from quaint English expressions

REPLY: We have reviewed the manuscript and removed any unnecessary expressions.

Reviewer 2 Report

Comments and Suggestions for Authors

I found this paper very interesting, addressing an area that is quite difficult to manage from a clinical point of view and relatively poorly investigated.

I also appreciate the accuracy of the writing, the quality of the scientific documentation, the logical systematization of the studied chapters, as well as the illustrative figures that, together, can represent a valuable resource for clinical practitioners.

I only suggest that some additional information about the ongoing RECORD study be provided (e.g. the investigator, the institution).

Author Response

I found this paper very interesting, addressing an area that is quite difficult to manage from a clinical point of view and relatively poorly investigated.

I also appreciate the accuracy of the writing, the quality of the scientific documentation, the logical systematization of the studied chapters, as well as the illustrative figures that, together, can represent a valuable resource for clinical practitioners.

I only suggest that some additional information about the ongoing RECORD study be provided (e.g. the investigator, the institution).

REPLY: Thank you for your kind comments.  We have now added the following based on your comments: “As an attempt to answer some of these questions, the our research team designed the RECORD (‘Rotation, Escalation, Combination, Or Reduction for Delirium) study to compare scheduled haloperidol, lorazepam, lorazepam plus haloperidol and placebo (NCT03743649) (Figure 1). This multicenter, double-blind, randomized trial enrolled 111 patients with advanced cancer from MD Anderson Cancer Center, Virginia Commonwealth University and Taiwan National University Hospital.  It has just completed enrollment and the results are expected in 2024.”

Reviewer 3 Report

Comments and Suggestions for Authors

The paper submitted by D. Hui and colleagues entitled “Pharmacologic management of End-of-Life Delirium: Translating Evidence to Practice“ offers a very complete review about the delirium treatment in terminal cancer patients. From my point of view the paper is suitable for publication as soon as authors take into account few considerations to make it more easy-reading. Above all because this is a very interesting review not only for clinicians, but also for nurses, pharmaceutics and other health staff.

- Authors should point out the role of no-pharmacological approaches, even if they are not the first option for the patient but they could be for the relatives.

- Criteria for patient inclusion in the palliative care units should be addressed.

- Please, even if they are well known mechanisms of action and clinical effects, it should be pointed out the use of Olanzapine, Haloperidol, Midazolam and so on in these palliative unit as well as if there is a protocol to use them in the above mentioned units.

- It is very important and significative the fact that in several studies there is not a placebo condition. Please explain that this could be no ethic and what is considered placebo in the papers mentioned in the review where there is this control condition.

- Bibliography is not in the correct order (line 36 for instance), please check it.

Author Response

The paper submitted by D. Hui and colleagues entitled “Pharmacologic management of End-of-Life Delirium: Translating Evidence to Practice“ offers a very complete review about the delirium treatment in terminal cancer patients. From my point of view the paper is suitable for publication as soon as authors take into account few considerations to make it more easy-reading. Above all because this is a very interesting review not only for clinicians, but also for nurses, pharmaceutics and other health staff.

 - Authors should point out the role of no-pharmacological approaches, even if they are not the first option for the patient but they could be for the relatives.

REPLY: Thank you for your insightful comments. Based on your feedback, we have now included the following: “The impact of non-pharmacologic measures on caregivers of patients with delirium, such as window light and music, remains a subject for further investigation. Education of caregivers of the nature history, manifestations and treatment options of end-of-life delirium, exploration of their values, and setting realistic expectations are essential.”

- Criteria for patient inclusion in the palliative care units should be addressed.

REPLY: Thank you for your insightful comments.  We have now added the following based on your feedback: “Much of the literature is derived from acute palliative care units, where patients are admitted for severe physical and/or psychosocial distress, need of intensive end-of-life care and/or complex discharge coordination. Although not typically a requirement, many of these patients have a prognosis in terms of days to weeks of survival and end-of-life delirium is common.”

- Please, even if they are well known mechanisms of action and clinical effects, it should be pointed out the use of Olanzapine, Haloperidol, Midazolam and so on in these palliative unit as well as if there is a protocol to use them in the above mentioned units.

REPLY: Thank you for your comment.  This review focused on existing literature on end-of-life delirium which is quite limited. Among the medications discussed were haloperidol, olanzapine, chlorpromazine, lorazepam and midazolam.  We have now revised the figure to list more options.

- It is very important and significative the fact that in several studies there is not a placebo condition. Please explain that this could be no ethic and what is considered placebo in the papers mentioned in the review where there is this control condition.

REPLY: Thank you for your comment.  We highlighted the issue with lack of placebo control in several studies as follows: (1) Breitbart: “However, the lack of a placebo control means that it was not possible to know if any improvement was as a result of neuroleptics or the natural history of delirium.” (2) Lin: “the small sample size and lack of a placebo control were important limitations.” (3) van der Vorst “this study also had several important limitations, including heterogenous patient populations, no masking of patients and clinicians, non-equivalent doses, lack of a placebo control and unclear survival.”

- Bibliography is not in the correct order (line 36 for instance), please check it.

REPLY: Thank you for noticing this.  We have now corrected this.

Round 2

Reviewer 1 Report

Comments and Suggestions for Authors

Thank you for the corrections and explanations for the reviewers' comments.

Essentially, you argue that the disease leading of delirium at the end of life is irrelevant and given the poor outcome, it may be irrelevant to investigate even remedial causes.